# Thermal Environment Map in Street Canyon for Implementing Extreme High Temperature Measures

**Hideki Takebayashi *** , **Mai Okubo and Hiroki Danno**

Department of Architecture, Graduate School of Engineering, Kobe University, Kobe 657-8501, Japan; 180t011t@stu.kobe-u.ac.jp (M.O.); 204t035t@stu.kobe-u.ac.jp (H.D.)
* Correspondence: thideki@kobe-u.ac.jp; Tel.: +81-78-803-6062

**Abstract:** The thermal environment map in street canyon is derived by using GIS building data and more detailed calculation, and its effectiveness is considered for implementing extreme high temperature measures. The influence of mean radiant temperature (MRT) is more dominant than the wind velocity on the distribution of standard new effective temperature (SET*) on the typical summer day in street canyon in the urban area of Kobe city, and the solar radiation shading is more effective in suppressing the rise of SET* in the daytime than improving the land coverage. The following strategy of extreme high temperature measures is derived by considering the thermal environment map in street canyon. Pedestrians may find the shaded places on the north-south road until 10:00 a.m. and after 3:00 p.m., due to the eastern building's shade in the morning and the western building's shade in the afternoon.

**Keywords:** adaptation measure; extreme high temperature; shading; land cover; street canyon

## 1. Introduction

Based on the experience of the extreme high temperature (heatwave) of the summer 2018, Kobe city has been studying and implementing specific measures to target extreme high temperature. A demonstration experiment of cool spots on the outdoor space was carried out from July 3 to September 30, 2019, where fractal sunshades with fine mists were installed on the plaza in front of the famous department store and the north-south street in front of Sannomiya station [1]. In response to a request from the Kobe city local government, the authors presented a proposal for Kobe city policy mainly on the basis of the knowledge of the art. It is constituted by priority introduction places for adaptation measures for extreme high temperature, effects of adaptation measures for extreme high temperature, and hot spots distribution in Kobe city. Kobe city held a symposium "Countermeasures against extreme high temperature" for citizens in July 2019 [2]. Therefore, the same expression is used in this manuscript.

As countermeasures against extreme high temperature, it is an urgent task to design a countermeasure strategy at the street canyon level. The Japanese Ministry of the Environment developed the "Heat countermeasure guideline in the city" [3], which includes basic, specific adaptation measures and technical sections. Several studies focused on effective measures against heat waves have been implemented in various countries [4–9]. Evaporative cooling effects such as irrigation [4,5], vegetation, and pavement watering [5] have been studied by the numerical simulation. Some of those scenarios assumed the future climate affected by climate change [5,6]. Discussions including the improvement of thermal environments in the street canyon or in the plaza were not sufficiently conducted based on the evaluation of the human thermal comfort in previous examinations [7–9]. According to a report from Karlsruhe city [10], it is recommended that appropriate adaptation measures

be introduced in "hot spots" where temperatures are high. Several typical urban districts in cities that may undergo adaptation in the future are also discussed.

In order to mitigate the negative impact by the extreme high temperature, various strategies have been developed [11], such as solar radiation shade, urban ventilation, and mist spray, among others. The appropriate strategy should be applied depending on the characteristics of each location. It follows that the urban climate map is an effective tool for the identification of places that need intervention and, at the same time, for the evaluation of which adaptation technique should be applied at each location. Many useful examples of large cities such as Tokyo and Beijing, medium-sized cities such as Salvador and Berlin, and small cities such as Sendai and Stuttgart were presented in the book published by Ng and Ren [12]. They contributed to bridging the gap between the science of urban climatology and the practice of urban planning. Many of the existing studies analyze urban climate maps at urban scale and are focused on air temperature and wind distribution in the entire urban area [13]. Differently, in order to focus on the radiation and wind effects on pedestrians, the analyzed area is at a district scale. Spatial distribution of air temperature and humidity are a little for human thermal environment in a street canyon. Effects on wind and radiant environment due to building and urban block characteristics should be clarified in order to produce reliable urban climate maps at district scale [3]. In this study, the thermal environment map in street canyon is derived by using GIS building data and more detailed calculation, and its effectiveness is considered for implementing extreme high temperature measures. The purpose of this study is to derive a strategy of extreme high temperature measures based on the thermal environment map in street canyon. The scale targeted in this study is smaller than the scale generally discussed as the local climate zone [14].

## 2. Methods and Results

Kobe city is located facing Osaka bay. The climate is classified as warm and temperate. According to Köppen and Geiger, this climate is classified as Cfa. The average annual temperature is 16.7 °C. The average annual rainfall is 1216 mm. Daily maximum and minimum air temperature in Kobe city, from July to September, 2017 to 2019, is shown in Figure 1. In Japan, a day with a minimum air temperature of 25 °C or higher is called a tropical night and used as an index of nighttime sleepiness, and a day with a maximum air temperature of 35 °C or higher is called an extreme hot day and is used as an index of daytime heat. The numbers of tropical nights from 2017 to 2019 were 52, 51, and 46 days, and an air conditioner was essential for sleeping. The numbers of extreme hot days from 2017 to 2019 were 3, 12, and 7 days, so the last 2 years were extremely hot.

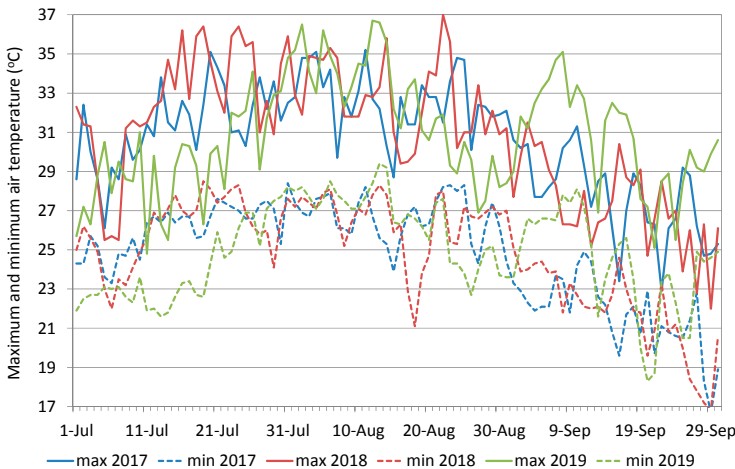

**Figure 1.** Daily maximum and minimum air temperature in Kobe city, from July to August, 2017 to 2019.

Incident solar radiation distribution was calculated by Arc-GIS tool, according to the method by the author's previous study [15]. In this method, direct and sky solar radiation influenced by

surrounding buildings were considered, but reflected solar radiation was not considered to avoid the calculation of complex inter-reflection of each surface. Since the urban blocks were not configured with surface materials with high reflectivity, the error due to this was considered to be small. Surface temperature was calculated on each ground and wall surface by the surface heat budget equation which balances radiation, convection, evaporation, and conduction heat flux, by given the physical properties of each surface [16]. Radiation heat flux was calculated by incident solar radiation and mutual infrared radiation by using building shape data. Convection and evaporation heat flux were calculated by air temperature, relative humidity, wind velocity, and surface temperature on each ground and wall surface, by using the heat and moisture transfer coefficients between the air and each surface. Conduction heat flux was calculated by the one-dimensional transient heat conduction equation with internal temperature as boundary condition on each surface material [16]. Since long-term calculations were not carried out, the effect of the stored heat of solar radiation in the morning on the surface temperature from afternoon to evening was considered, but the effect of the previous day's effect on the surface temperature in the morning was not considered.

## 2.1. Wind Velocity Distribution

Calculation area which is an outer square with 2.3 km × 2.3 km (Area 1) and analysis area which is an inner square with 1.0 km × 1.3 km (Area 2) are shown in Figure 2. A buffer area was set outside this so that the setting of the peripheral boundary conditions did not affect the calculation results. Most frequent wind direction was south-west and averaged wind velocity was 4.3 m/s during 1–31 August 2010 at 100 m height on Port-tower observatory which is located on southwest, that is, on the windward side of the analysis area. Since the frequency of land breeze decreased due to urbanization, a land breeze with opposite wind direction to sea breeze was sometimes confirmed at night, but the wind direction was almost stable due to the sea breeze during the day. Therefore, these values were used to set the initial and inflow boundary conditions. Vertical wind profile was given assuming a typical boundary layer in the urban area. The standard k-ε model was used for the turbulence closure model. The altitude was assumed to be constant in the calculation area. Although railway paths and overpasses were not reproduced, the building shapes were reproduced. The calculation result of wind velocity distribution at 1.5 m height is shown in Figure 3. Wind velocity was large in the wider street and open space, but was small in the street canyon due to the surrounding buildings. Although 4.3 m/s was given to the upper wind velocity, the wind velocity on the pedestrian level was high only in the windward waterfront area and was less than 1 m/s in 90.6 % of the objective area, due to the resistance by the middle-rise buildings.

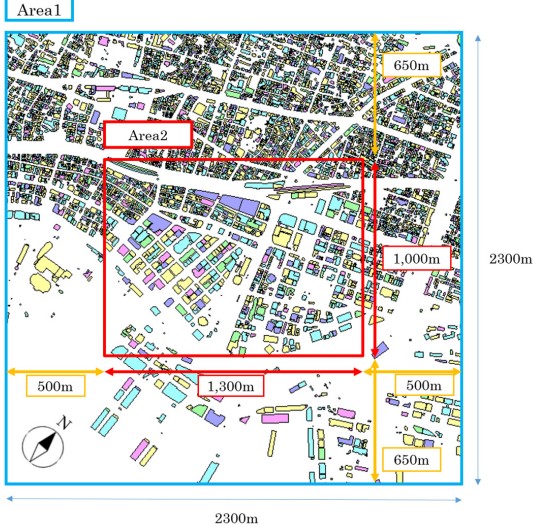

**Figure 2.** Calculation area (Area 1) and analysis area (Area 2).

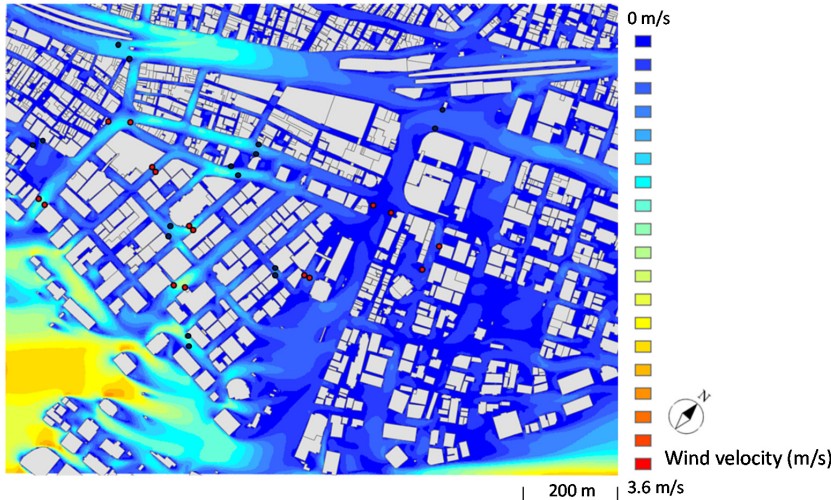

**Figure 3.** Calculation result of wind velocity distribution at 1.5 m height in analysis area. Red circles (16 pairs) indicate measurement points.

### 2.2. Thermal Sensation Index Distribution

As Nouri et al. [17] pointed out, the selection of the index for the assessment of outdoor thermal comfort conditions is still a debated matter [18]. They stated that "So far, within the international community various indices have been developed and disseminated, including the (i) Standard Effective Temperature (SET*) [19]; (ii) Outdoor Standard Effective Temperature (OUT_SET*) [20,21]; (iii) Perceived Temperature (PT) [22]; (iv) Predicted Mean Vote (PMV) [23,24]; (v) Index of Thermal Stress (ITS) [25]; (vi) Predicted Percentage of Dissatisfied (PPD) [24]; (vii) COMFA outdoor thermal comfort model [26]; (viii) Universal Thermal Climate Index (UTCI) [27–30]; (ix) Wet Bulb Globe Temperature (WBGT) [30,31]; and (x) Predicted Heat Strain (PHS) [32–34]." In Japan, SET* and WBGT are mainly used. WBGT, which is a stress index worldwide accepted as a preliminary tool for the assessment of hot thermal environments [35–37], is often used under more severe conditions to warn of the risk of heat stroke. SET* is defined as the equivalent dry bulb temperature of an isothermal environment at 50% RH in which a subject, while wearing clothing standardized for the activity concerned, would have the same heat stress and thermoregulatory strain as in the actual test environment [19], is used to evaluate the thermal environment [3]. The relationship between SET* and thermal comfort is associated based on the results of a declaration test for the outdoor comfort of Japanese people [38]. SET* is desirable as an index from the viewpoint of appropriately introducing adaptation measures in urban areas and developing a more comfortable outdoor space as it exhibits a good relationship with outdoor thermal comfort [39].

Mean radiant temperature (MRT) was calculated by incident solar and infrared radiation to human body, which was calculated by surrounding surface temperature and view factors between the human body and surrounding surfaces. The view factor between each surface and the human body was calculated by assuming the human body as a sphere. SET* was calculated by integrating the wind velocity and MRT distributions, by giving air temperature, relative humidity, a clothing amount, and a metabolic rate of the human body. Calculations were performed on GIS using a program that was modified and ported by the authors.

Building shape and land cover distribution in the objective area, classification of open space in the study area, area ratio of land cover materials in each study area, distribution of trees, ratio of tree canopy area in each study area, properties of land cover materials are shown in Figures 4–7 and Tables 1 and 2, respectively. The objective area is the economic, administrative, and cultural center of Kobe city, where people gather from inside and outside the city. Sannomiya station is located in the northeastern end and Motomachi station is located in the northwestern end of this area. The south end

of this area is connected to the port area. The east-west shopping streets are located on the north side of this area, but because these pedestrian streets have arcades, they are excluded from the calculation in this study. Relatively large-scale buildings such as offices, department stores, banks, museums, hotels, city halls, and condominiums are located from the center to the south side of this area. The objective area was divided into 2 m grid, and surface materials were set for each grid. Although asphalt, block, and grass were set for the surfaces in the street canyons, wood deck, grass, water surface, asphalt, concrete, soil, block (white), block (colored), brick, andesite were set for the surfaces only in the central park (Higashi-Yuenchi). The crown width and tree height of each street tree and park tree were set by a field survey and Google Earth. Ratio of tree canopy area in the entire objective area was 9.7%, which was almost the same in the east-west road and the north-south road; it was large at 31.9% in central park and small in intersection and open space. The accuracy of the calculation result is compared with the measurement result in Section 3.3.

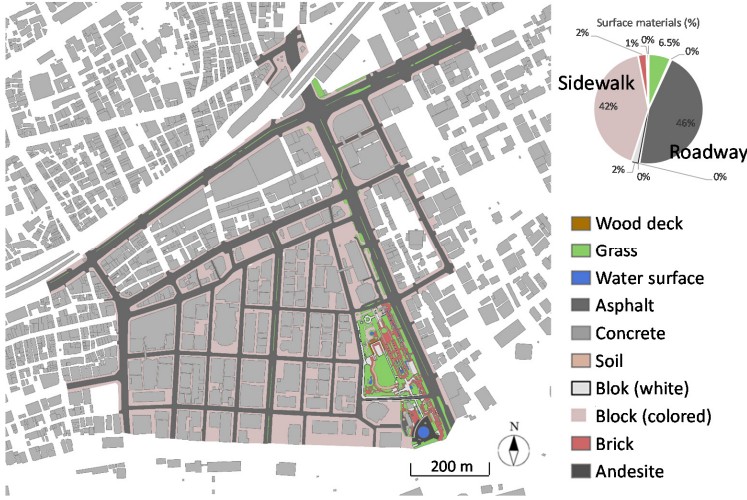

**Figure 4.** Building shape (gray—top view) and land cover distribution in the objective area.

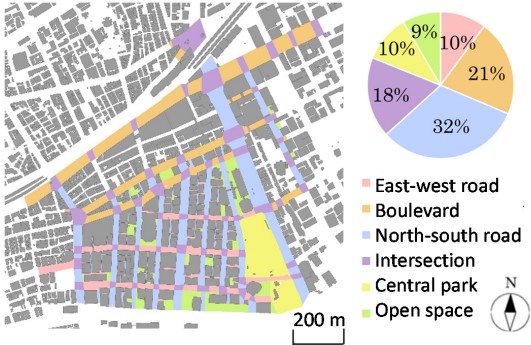

**Figure 5.** Classification of open space in the study area.

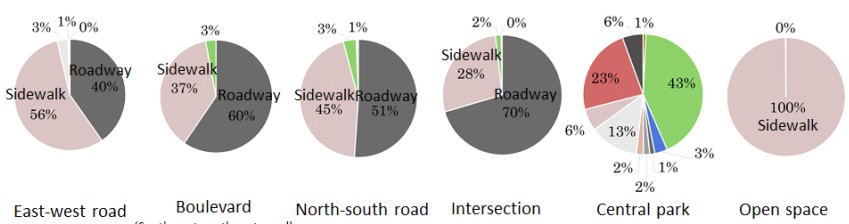

**Figure 6.** Area ratio of land cover materials in each study area.

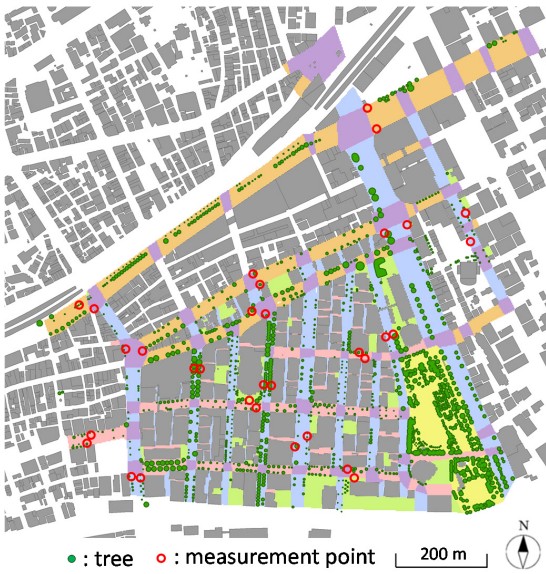

**Figure 7.** Distribution of trees in the study area.

**Table 1.** Ratio of tree canopy area in each study area.

| Area | Ratio of Tree Canopy Area |
|---|---|
| East-west road | 9.5% |
| Boulevard | 7.1% |
| North-south road | 9.4% |
| Intersection | 3.3% |
| Central park | 31.9% |
| Open space | 4.7% |
| All area | 9.7% |

**Table 2.** Properties of land cover materials.

| Material | Evaporative Efficiency (-) | Emissivity (-) | Thermal Conductivity (W/(mk)) | Heat Capacity (kJ/(m$^3$K)) | Solar Reflectance (-) |
|---|---|---|---|---|---|
| Wood deck | 0.0 | 1.0 | 0.5 | 1130 | 0.15 |
| Grass | 0.3 | 0.9 | 3.0 | 3000 | 0.3 |
| Water surface | 1.0 | 1.0 | 7.0 | 9000 | Depends on incident angle |
| Asphalt | 0.0 | 1.0 | 0.74 | 2055 | 0.15 |
| Concrete | 0.0 | 0.95 | 1.7 | 1934 | 0.35 |
| Soil | 0.0 | 0.92 | 0.74 | 10,000 | 0.225 |
| Block (white) | 0.0 | 0.9 | 1.4 | 2000 | 0.39 |
| Block (colored) | 0.0 | 0.9 | 1.4 | 2000 | 0.3 |
| Brick | 0.0 | 0.9 | 0.62 | 1400 | 0.15 |
| Andesite | 0.0 | 0.9 | 1.7 | 3030 | 0.15 |

## 3. Results

### 3.1. Calculation Results

Time change of the ground surface temperature, MRT, and SET* are calculated on a typical summer sunny day, 5 August 2019. Air temperature and relative humidity are given the measurement data by Kobe local meteorological observatory located nearby the objective area. It is assumed that a solar absorption rate, a clothing amount, and a metabolic rate of the human body are 0.5, 0.6 clo, and 1.0 met,

respectively, and transmittance of solar radiation of the tree is 0.06. Metabolic rate of the human body should be set according to the activity, e.g., walking on roads, standing at intersections, sitting in open spaces, walking, sitting, and exercising in parks. If these values are respectively set according to each place, recognition of the SET* distribution becomes complicated, so a constant value of 1.0 met when the human body is at rest is given throughout the objective area.

### 3.2. Distribution of Ground Surface Temperature, MRT and SET*

Distribution of ground surface temperature, MRT, and SET* at 1.5 m height at 1:00 p.m. on 5 August 2019 is shown in Figures 8 and 9. Ground surface temperature is low in the shaded area by the surrounding buildings and trees. Since daily integral incident solar radiation is small in the shaded streets by street trees, surface temperature is low in both east-west and north-south roads. MRT is low in the median strip in some streets and the central park, where incident solar radiation and surrounding surface temperature are low. SET* is more affected by MRT than wind velocity.

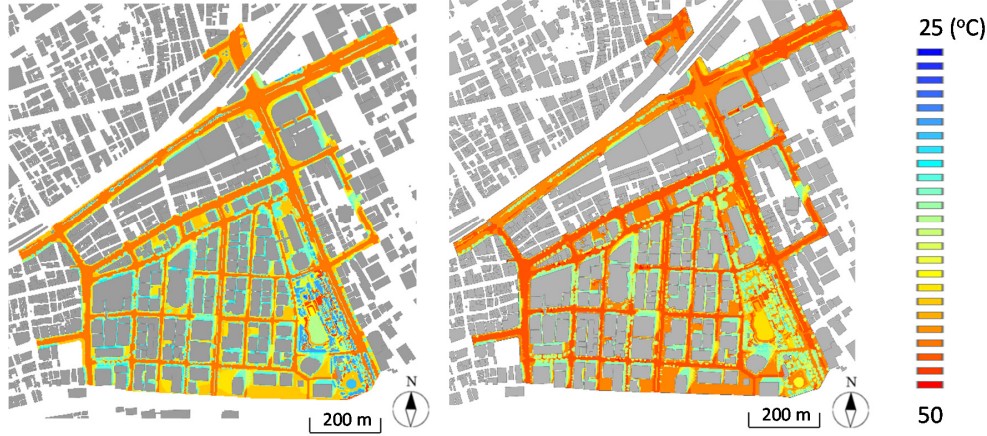

**Figure 8.** Distribution of ground surface temperature (left) and mean radiant temperature (MRT) (right) at 1.5 m height at 1:00 p.m. on 5 August 2019.

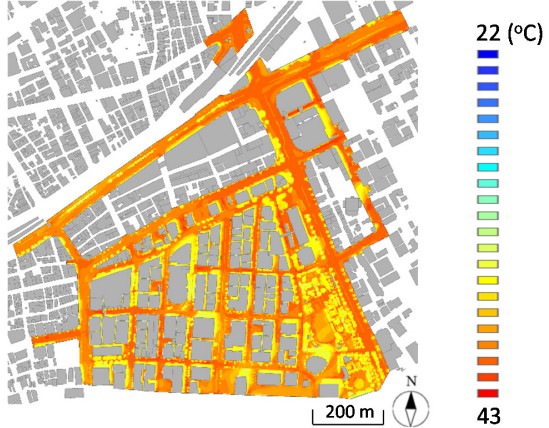

**Figure 9.** Distribution of Standard Effective Temperature (SET*) at 1.5 m height at 1:00 p.m. on 5 August 2019.

### 3.3. Comparison with Measurement Results

Air temperature, relative humidity, wind velocity, and surface temperature on the east-west and north-south roads with various street widths in the objective area were measured from 11:00 a.m. to 1:00 p.m. and from 3:00 p.m. to 5:00 p.m., on 29 July 7 August and 24 September 2019. Measurement points are shown in red circles in Figure 3. Altogether, 16 pairs of measurement points are set on

the sidewalks on both sides of the east-west and north-south streets with different road widths. Two persons moved each measurement point during the above period and measured them after staying at each measurement point for a while. Air temperature and relative humidity were measured at 1.5 m height by a thermistor and a capacitance sensor inside a ventilation system with a shade. Wind velocity was measured by a hot-wire anemometer after confirming the wind direction by a windsock. Surface temperature on the surrounding ground and wall surfaces was measured by an infrared thermometer, which was supplemented by an infrared image by a thermal camera. A fisheye photograph was taken at each measurement point, then MRT and SET* were calculated, given the same assumptions as the above calculation methods. Measurement results of MRT and SET* at 1.5 m height around noon on 7 August 2019 are shown in Figure 10. Since the measurement results on 29 July and 7 August were almost the same, it is recognized to be the results in the typical summer sunny day in the year 2019. There are several measurement points with lower MRT and SET* on the south side of east-west road. Lower MRT and SET* are also confirmed at the measurement points where the roadside trees make shade. Shading results in lower MRT and lower SET*. There are maximum differences of about 20° in MRT and 10° in SET*, which are consistent with the tendency of the calculation results indicated later in Figures 13–15. Since it took about 2 h for mobile measurements, the spatial difference was indistinguishable from the temporal difference in air temperature and relative humidity measurement results. On the other hand, a clear difference was confirmed in the spatial distribution of MRT based on the sun shade and the surface temperature distribution. Although slight difference was also confirmed in the spatial distribution of wind velocity affected by surrounding features, the influence of the MRT was dominant for the spatial distribution of SET*.

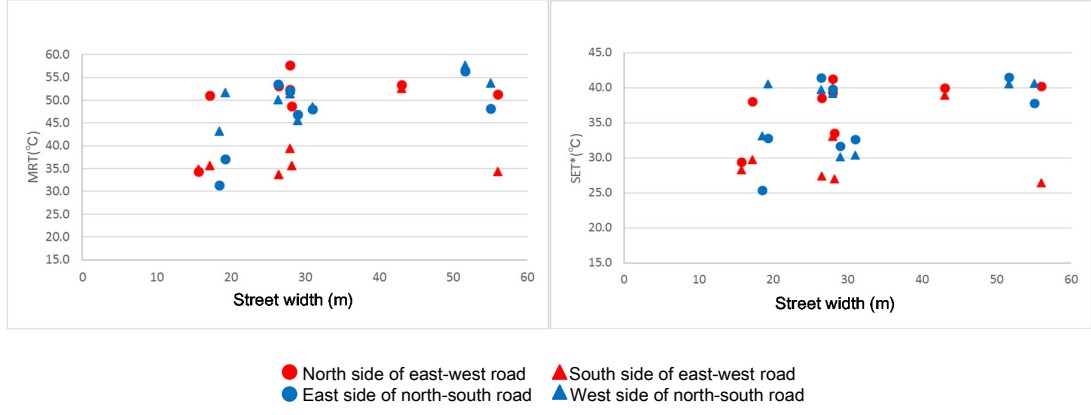

**Figure 10.** Measurement results of MRT (left) and SET* (right) at 1.5 m height around noon on 7 August 2019.

## 4. Discussion

### 4.1. Time Change of SET*

Diurnal variation of SET* at 1.5 m height on each land cover on 5 August 2019 is shown in Figure 11. The averaged value and the standard deviation are calculated for each sunny and shaded place on each cover, from the calculation results of the spatial distribution of SET*. The relationships between SET* and thermal sensation by Ishii et al. [38] are shown as horizontal lines. Based on these relationships, it is comfortable on all land covers from 6:00 a.m. to 7:00 p.m., but is very uncomfortable in the sunny place except on grass and water surface after 9:00 a.m. It is uncomfortable even in the shaded place on grass and water surface from 1:00 p.m. to 2:00 p.m. It is neutral in the shaded place on water surface at 6:00 p.m. Solar radiation shading is more effective in suppressing the rise of SET* than the difference of land cover.

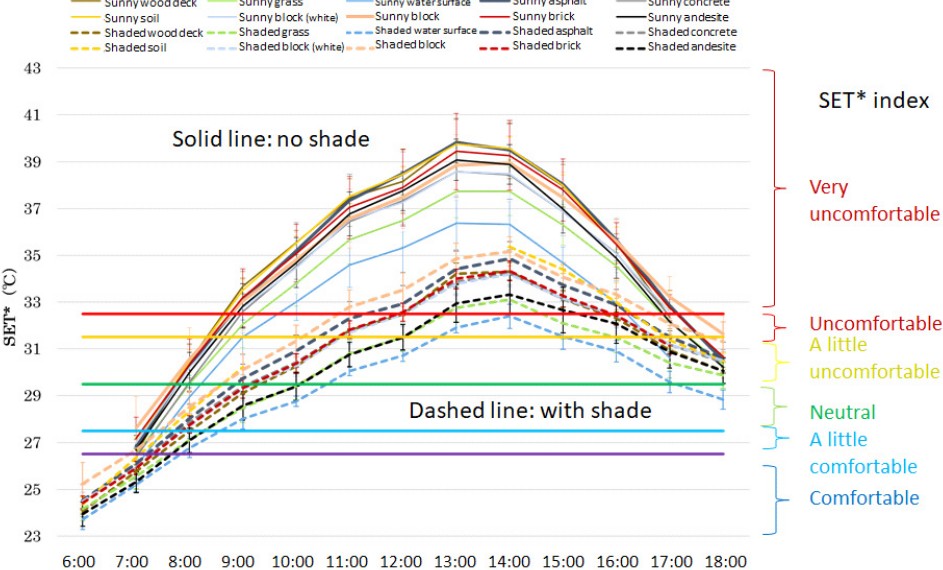

**Figure 11.** Diurnal variation of modeled SET* at 1.5 m height on each land cover on 5 August 2019.

*4.2. Spatial Distribution Frequency of SET* and Shaded Area*

Diurnal variation of the spatial distribution frequency of SET* at 1.5 m height in each study area on 5 August 2019 is shown in Figure 12. SET* is low in shaded area since incident solar radiation is dominant on SET*. Diurnal variation of the shaded area ratio in each study area on 5 August 2019 is shown in Figure 13. Although the proportion of the shaded area is steady at 30% to 40% from 9:00 a.m. to 4:00 p.m. on the east-west road and in the central park, it becomes the minimum around noon in other areas. Even around the noon, the pedestrians may find the shaded places on the east-west road due to the southern building's shade and in the central park due to the tree's shade. On the other hand, pedestrians may find the shaded places on the north-south road until 10:00 a.m. and after 3:00 p.m., due to the eastern building's shade in the morning and the western building's shade in the afternoon. Overall, pedestrians should look for places to avoid extreme heat on the north-south road in the morning and afternoon and on the east-west road and in the central park around the noon. It is difficult to find shaded places at intersections, open spaces, and boulevards (southwest-northeast road).

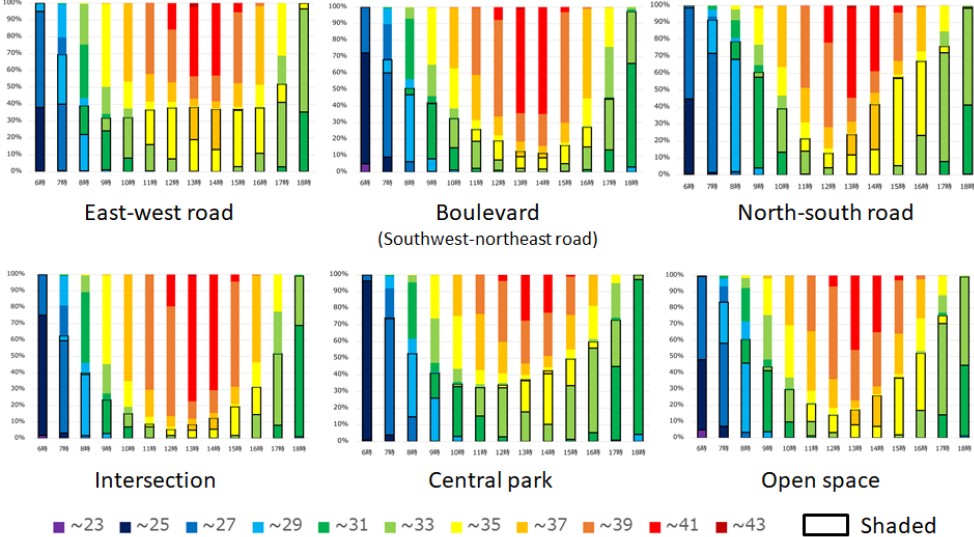

**Figure 12.** Diurnal variation of the spatial distribution frequency of modeled SET* at 1.5 m height in each study area on 5 August 2019.

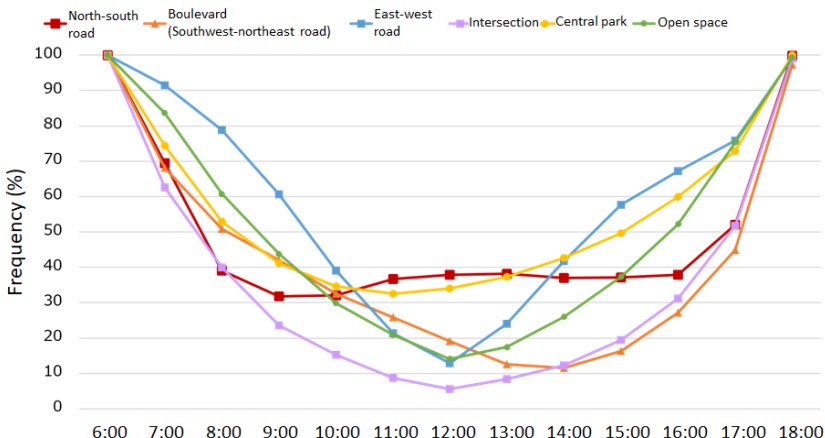

**Figure 13.** Diurnal variation of the modeled shaded area ratio in each study area on 5 August 2019.

### 4.3. Relationship between Street Width and SET*

Relationship between street width and SET* at 1.5 m height at 1:00 p.m. on 5 August 2019 on east-west road, boulevard (southwest-northeast road), and north-south road is shown in Figures 14–16 Asphalt generally corresponds to the road, and block generally corresponds to the sidewalk. When the street width becomes narrow, SET* tends to decrease slightly in the sunny place and increase slightly in the shaded place. Sunny places are generally warmer (hotter) than shaded places in all widths of street canyons. In all three roads, if the street is divided into sunny and shade places, the variation of SET* is small because the influence of solar radiation is large.

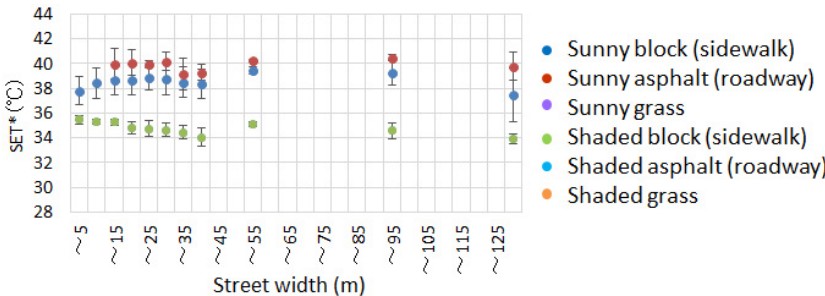

**Figure 14.** Relationship between street width and SET* at 1.5 m height at 1:00 p.m. on 5 August 2019 on east-west road. The plot is the averaged value, and the vertical bar on each plot is the standard deviation in the street width section.

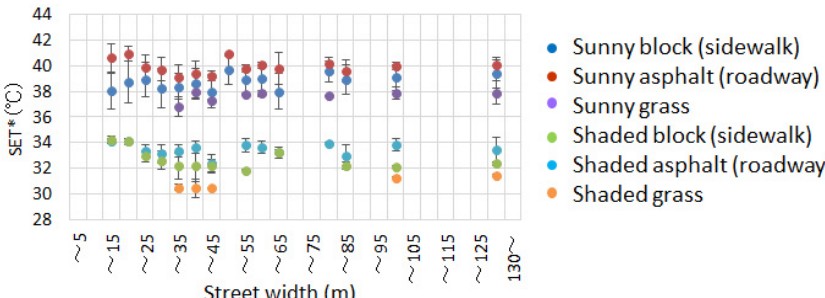

**Figure 15.** Relationship between street width and SET* at 1.5 m height at 1:00 p.m. on 5 August 2019 on boulevard (southwest-northeast road). The plot is the averaged value, and the vertical bar on each plot is the standard deviation in the street width section.

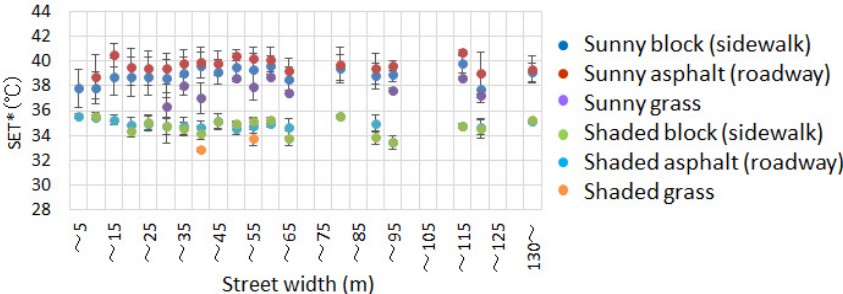

**Figure 16.** Relationship between street width and SET* at 1.5 m height at 1:00 p.m. on 5 August 2019 on north-south road. The plot is the averaged value, and the vertical bar on each plot is the standard deviation in the street width section.

## 5. Conclusions

In this study, it was considered to derive a strategy of extreme high temperature measures based on the thermal environment map in street canyon. The influence of MRT is more dominant than the wind velocity on the distribution of SET* on the typical summer day in the urban area of Kobe city, and the solar radiation shading is effective in suppressing the rise of SET* in the daytime than improving the land coverage. However, this paper presents the analysis results only on a specific typical sunny summer day. It represents a period called the extreme hot day in Japan, but additional analysis is needed for cloudy days and days with different solar altitudes. Based on the relationships between SET* in the daytime and thermal sensation, there is no place in all objective areas that people feel comfortable, and the proportion of places that people feel extremely uncomfortable is large even in the shaded places. The percentage of shaded areas varies from 30% to 40% from 9:00 a.m. to 4:00 p.m. both on the east-west road due to the southern building's shade and in the central park due to the tree's shade, but it becomes the minimum value around the noon on other roads. Even around the noon, the pedestrians may find the shaded places on the east-west road due to the southern building's shade and in the central park due to the tree's shade. On the other hand, pedestrians may find the shaded places on the north-south road until 10:00 a.m. and after 3:00 p.m., due to the eastern building's shade in the morning and the western building's shade in the afternoon. Pedestrians should look for places to avoid extreme heat on the north-south road in the morning and afternoon and on the east-west road and in the central park around the noon.

**Author Contributions:** Conceptualization, H.T.; methodology, H.T.; software, M.O.; validation, M.O., H.D., and H.T.; formal analysis, M.O. and H.D.; investigation, H.T.; resources, M.O. and H.D.; data curation, M.O. and H.D.; writing—original draft preparation, M.O. and H.D.; writing—review and editing, H.T.; visualization, M.O.; supervision, H.T.; project administration, H.T.; and funding acquisition, H.T. All authors have read and agreed to the published version of the manuscript.

**Funding:** This research was supported by Kobe city.

**Acknowledgments:** This research was conducted in cooperation with Kobe city. The authors thank Ushio Tozawa and Toshihiko Inamatsu of Kobe city.

**Conflicts of Interest:** The authors declare no conflicts of interest.

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
