# Peer review of "Thermal Environment Map in Street Canyon for Implementing Extreme High Temperature Measures"

_atmosphere, doi:10.3390/atmos11060550_

Round 1

Reviewer 1 Report

  1. Please clarify the objective of this study. Is it the development of thermal environment map, or the assessment of pedestrian comfort?
  2. Please specify the posture of the human body for calculating MRT.
  3. 1.0 met is a typical metabolic rate for a seated sedentary person, which seems inappropriate for pedestrians in streets. The validity of the assumption is closely related to the objective of this study. Please explain why metabolic rate of 1.0 met was assumed for this study.

Author Response

Thank you for reviewing our manuscript. We respond to your comments as follows.

Point 1: Please clarify the objective of this study. Is it the development of thermal environment map, or the assessment of pedestrian comfort?

Response 1: We added "The purpose of this study is to derive a strategy of extreme high temperature measures based on the thermal environment map in street canyon." (Lines 60, 61).

Point 2: Please specify the posture of the human body for calculating MRT.

Response 2: We added "The view factor between each surface and the human body is calculated by assuming the human body as a sphere.” (Lines 115, 116)

Point 3: 1.0 met is a typical metabolic rate for a seated sedentary person, which seems inappropriate for pedestrians in streets. The validity of the assumption is closely related to the objective of this study. Please explain why metabolic rate of 1.0 met was assumed for this study.

Response 3: We added "Metabolic rate of the human body should be set according to the activity, for example, walking on roads, standing at intersections, sitting in open spaces, walking, sitting, and exercising in parks. If these values are respectively set according to each place, recognition of the SET* distribution becomes complicated, so a constant value of 1.0 met when the human body is at rest is given throughout the objective area.” (Lines 146-150)

Reviewer 2 Report

This paper tries to reveal a thermal environmental map that can be used for urban pedestrians. However, the findings from the measured data and simulation are quite general (common sense), which makes this paper unclear for the title.
Overall it is a very interesting research topic, but there are several issues that need to be addressed in the manuscript, and I recommend that the authors should further improve their proposed approach as shown below.

My comments are:
1. In the abstract, the contribution of the findings needs to be addressed.

2. Need to add more literature reviews in the introduction section. A general reference to previous researches and activities regarding outdoor thermal comfort which would lead to show why SET* is chosen for this research.

3. Line 48-50 requires to reference for the sentences.

4. The definition of Extreme high-temperature measures needs to be clarified. Does it represent a special measurement? Author needs to address why using the term. Is there any special methodology to measure extremely high temperature?

5. Line 62. What are the physical properties of each surface? If all materials applied in the simulation, it needs to be mentioned though building materials on the urban scale prefer to be simplified.

6. Line 66-67. What equation? Any reference?

7. Line 89. What software is used for the simulation? If it is explained in the introduction as a literature review would be fine.

8. Line 94-95. Is there any calibration between field measure and simulation for the data reliability of the simulation?

9. Line 111. Thermal comfort calculated is preferable to be illustrated as a section drawing which can help to understand the mean thermal comfort.

10. It is not clear what the critical findings are in the conclusion section as simple general findings are just explained. Is this paper trying to suggest a new (unique) methodology or any critical findings in the urban context?

Author Response

Thank you for reviewing our manuscript.

This paper tries to reveal a thermal environmental map that can be used for urban pedestrians. However, the findings from the measured data and simulation are quite general (common sense), which makes this paper unclear for the title.

Response: In order to properly express the purpose of this study, we added "The purpose of this study is to derive a strategy of extreme high temperature measures based on the thermal environment map in street canyon." (Lines 60, 61).

Overall it is a very interesting research topic, but there are several issues that need to be addressed in the manuscript, and I recommend that the authors should further improve their proposed approach as shown below.

Response: Thank you for your detailed comments. We respond to your comments as follows.

My comments are:

Point 1: In the abstract, the contribution of the findings needs to be addressed.

Response 1: We added "The following strategy of extreme high temperature measures is derived by considering the thermal environment map in street canyon.” (Lines 15, 16)

Point 2: Need to add more literature reviews in the introduction section. A general reference to previous researches and activities regarding outdoor thermal comfort which would lead to show why SET* is chosen for this research.

Response 2: As a general reference, we added "Evaporative cooling effects such as irrigation [4,5], vegetation and pavement watering [5] have been studied by the numerical simulation. Some of those scenarios assumed the future climate affected by climate change [5,6]. Discussions including the improvement of thermal environments in the street canyon or in the plaza were not sufficiently conducted based on the evaluation of the human thermal comfort in previous examinations [7–9].” (Lines 39-44).

And, as a reference on outdoor thermal comfort, we added "As Nouri et al. [16] pointed out, the selection of the index for the assessment of outdoor thermal comfort conditions is still a debated matter [17]. They stated that, “So far, within the international community various indices have been developed and disseminated, including the (i) Standard Effective Temperature (SET*) [18]; (ii) Outdoor Standard Effective Temperature (OUT_SET*) [19,20]; (iii) Perceived Temperature (PT) [21]; (iv) Predicted Mean Vote (PMV) [22,23]; (v) Index of Thermal Stress (ITS) [24]; (vi) Predicted Percentage of Dissatisfied (PPD) [22]; (vii) COMFA outdoor thermal comfort model [25]; (viii) Universal Thermal Climate Index (UTCI) [26–29]; (ix) Wet Bulb Globe Temperature (WBGT) [29,30]; and (x) Predicted Heat Strain (PHS) [31–33].” (Lines 95-102).

And, regarding the reason why SET* is adopted, we added "In Japan, SET* and WBGT are mainly used. WBGT, which is a stress index worldwide accepted as a preliminary tool for the assessment of hot thermal environments [34–36], is often used under more severe conditions to warn of the risk of heat stroke. SET* is defined as the equivalent dry bulb temperature of an isothermal environment at 50% RH in which a subject, while wearing clothing standardized for the activity concerned, would have the same heat stress and thermo-regulatory strain as in the actual test environment [18], is used to evaluate the thermal environment [3]. The relationship between SET* and thermal comfort is associated based on the results of a declaration test for the outdoor comfort of Japanese people [37]. SET* is desirable as an index from the viewpoint of appropriately introducing adaptation measures in urban areas and developing a more comfortable outdoor space as it exhibits a good relationship with outdoor thermal comfort [38].” (Lines 103-112).

Point 3: Line 48-50 requires to reference for the sentences.

Response 3: We added reference number [3] (Line 58)

Point 4: The definition of Extreme high-temperature measures needs to be clarified. Does it represent a special measurement? Author needs to address why using the term. Is there any special methodology to measure extremely high temperature?

Response 4: We added "Kobe City held a symposium "Countermeasures against extreme high temperature" for citizens in July 2019 [2]. Therefore, the same expression is used in this manuscript.” (Lines 32-34)

Point 5: Line 62. What are the physical properties of each surface? If all materials applied in the simulation, it needs to be mentioned though building materials on the urban scale prefer to be simplified.

Response 5: We added Table 2 Properties of land cover materials (Line 139)

Point 6: Line 66-67. What equation? Any reference?

Response 6: We added reference number [15] (Line 75)

Point 7: Line 89. What software is used for the simulation? If it is explained in the introduction as a literature review would be fine.

Response 7: We added "Calculations are performed on GIS using a program that is modified and ported by the authors." (Lines 118, 119)

Point 8: Line 94-95. Is there any calibration between field measure and simulation for the data reliability of the simulation?

Response 8: We added "The accuracy of the calculation result is compared with the measurement result in Section 3.3.” (Lines 127, 128)

Point 9: Line 111. Thermal comfort calculated is preferable to be illustrated as a section drawing which can help to understand the mean thermal comfort.

Response 9: We added a new section 3.2 Distribution of ground surface temperature, MRT and SET* (Line 151)

Point 10: It is not clear what the critical findings are in the conclusion section as simple general findings are just explained. Is this paper trying to suggest a new (unique) methodology or any critical findings in the urban context?

Response 10: We added to the conclusion "In this study, it was considered to derive a strategy of extreme high temperature measures based on the thermal environment map in street canyon.” (Lines 234, 235)

Reviewer 3 Report

To the authors.

First of all, I want to point out that I appreciate the work and research you did. I was asked to review your manuscript. Due to the number of comments, I chose this chronicle way through the text referring to line numbers, figures and tables. Keep in mind that even enhancing the text by reflecting my comments might lead to new questions and comments.

Comments on the ABSTRACT

Line 12 and 13:

What MRT and SET* stand for? There are NO definitions of this in the whole article. I assumed that you meant mean radiant temperature, abbreviated MRT, or in another version, Tmrt. Probably, with SET* you refer to the Standard Effective Temperature. Please correct this and explain furthermore, why you used SET* instead other thermal comfort indices. Don't forget the references.

Line 14/15:

Please add 'more' to '…shading is more effective in…' and 'the' to 'the daytime than the improving the land coverage'.

Further comments on the Abstract

You should here describe your results in a more common way. To my opinion the sentence starting from 'Even around noon, the pedestrians … ' do not belong here.

Comments on the INTRODUCTION

First of all, I was expecting more information about the thermal conditions of the city of Kobe. By reviewing your research somewhere from the opposite side of Earth, I know less about the climate and the city of Kobe itself. Actually, there was no definition of extreme temperature. Thermal perception also varies in different climates. To understand the thermal severity it might be useful to have information about heatwaves (occurrence and length). Please add some basic climatological information. Also, a short geographical description of Kobe is needed. Kobe, as a city located on sea-side, benefits from the maritime aspects of the climate (sea breeze, humidity). Therefore it is interesting that you found out that in Kobe solar insolation is more dominant on the distribution of SET* as the wind.

Another aspect to understand your research is why this area was chosen as an area of interest (AOI). It seems that the building structure is two-folded, in the westerly part of AOI are according to maps are middle to large size buildings and in the easterly part is smaller to middle size houses. In such a type of research, you performed the contracting authorities normally want to know more about the thermal influences on different types of inhabitants or customers of the AOI. This increases the impact of researches on decision-makers. Are there such aspects of research? If there are, mention them and refer to other studies with references. 

One word about the term 'land cover (also land coverage)', mentioned, inter alia, in the keywords and line 15. To my understanding, there are a number of researches about the influences of the different land cover pattern on urban micro-climate.  Normally they refer to a longer period of investigation under different types of weather. But it is still very difficult to extract the impact of a specific small land cover, especially, if you have only one day with validation measurement. Later in my comments about your validation measurements, I will open-up the issue more.

Line 27:

The sentence starting in line 27: 'In response to a request …' is very long and at the end quite unclear. Please rephrase it.

Line 36:

To enhance the relevance of your scientific work you must substantially increase the number of references and refer to them one by one. Here you mentioned six references from ONE issue. This is not enough. Because there is already plenty of research around the world concerning the research topic, you will find easily different opinions and solution.

Line 45/46:

Here again, you have to open-up the references. Reference [13] is a review article - not research on urban climate maps.

Line 48:

What do you mean with the sentence: 'Spatial distribution …' and especially with the word 'little'?

At the end of the introduction, you should mention at least the definition of the Local Climate Zones by Stewart, I.D. and Oke, T.R. (2012).

At least here at the end of the introduction you should define and rephrase your research goals.

Comments on chapter 2

Make the header to 'Methods and results', skip 'Calculation'.

Open-up the SET* thermal index: development, history, use, advantages and possible disadvantages.

Line 56 - 58:

This is a good remark of direct solar radiation. But buildings can store energy in the late forenoon hours and radiate it in the early afternoon hours. You might mention this option, too.

Line 69:

You were analysing day-time weather condition. Are the wind conditions during day and night the same? Kobe is located on the sea-site. No stronger winds were measured? What about land - and sea-breeze circulations?

Line 78:

On line 78 you used the term '1.5 m high' for the first time. In thermal map research, you are focused on a specific height, here 1.5 metres. This height could stand for the centre of the human corpus. Other thermal indices like PET refer to 1.75 metres as height and therefore as a parameter to calculate the body surface. Please switch the indication of size from 'high' to 'height'.

Figure 2:

Compass rose missing. Legend missing. There are 8 pairs of measurement points (red dots), what are the other green dots standing for?

The wind floating scale of 18 classes with wind maximum of 3.6 metres/second out-site the AOI is nice but not very effective, you cannot see any differences.  

Line 84/85:

Red circles: You did not mention how many measurement points you had, how did you choose them, and where they were located.

Line 91:

What is the definition of 'Building shape'?

Line 94:

We normally use 'grid' instead of 'mesh'. Correct this from here on.

Line 97:

How many trees are in the AOI?

Figure 3:

Compass rose missing (important, because the figure is slightly turned). Typo in the legend: Block (white).

Figure 4:

Compass rose missing (important, because the figure is slightly turned).

I suggest a different title: Classification of open space in the study area.

Figure 6:

Compass rose missing (important, because the figure is slightly turned).

Legend missing.

And again an addition to the title: trees in the study area. To support the measurements results, it would be nice to indicate the sites of the measurements (red dots) here, too.

Comments on chapter 3

Line 112 - 114:

If you decide to define SET*, then this sentence is not necessary any more.

Figure 7 and 8:

Compass rose missing (important, because the figure is slightly turned).

As mentioned you have 8 measurement points (!), you did not mention the measuring height at these dots. Furthermore, did you use recording devices for measurements? Where the devices hand-held or stand-alone? How many thermal pictures were taken and how? Unfortunately, you don't have and measurements in the open park area, which make references of SET* and MRT to this are not proved.

Line 138 - 140:

Add 'in the year 2019' to the end of the sentence, because you cannot generalised from two days measurements, that you have a typical overall summer day.

Figure 9:

Generally speaking, you were measuring neither MRT nor SET*, both are calculated from air temperature, humidity, radiation and wind, as you should describe earlier. The MRT and SET* interrelationship is well seen. But as you are interested in extreme high temperature an additional figure with these really measured air temperatures would be here a useful complement.

Comments on chapter 4

You should mention here that the discussion is based on modelled data.

Line 158:

What do you mean by 'improving the land coverage'? Isn't the increase of shading (e.g. by trees or other obstacles) an improving of the land coverage? I don't see here the controversial aspect.

Figure 10, 11, 12:

Add ' Modelled' to the title.

Line 185 - 186:

How this dominance is expressed? Is it not possible just to say that sunny places are generally warmer (hotter) than shaded places in all widths of street canyons?

Figure 13 - 15:

What are the whiskers in the figures stands for? Are they needed?

How do explain the low variability of SET* of all street canyon types and widths? For sunny and shaded sites the differences are within 2 degrees of SET* and still in the discomfort zone of SET*.

Comments on chapter 5

Line 197:

Because you modelled SET* and MRT on one day and measured the air temperature, humidity, thermal building conditions, and wind twice in two-hour sessions for the selected sites on that day, this is not enough to classify this day as typical summer day without any validation with long-term climate data. Please add this as a notice, that your research is based on day, August, 5th, 2019, or validate your measurements against climate data.

Then you can conclude that MRT is more dominant than wind velocity on SET*.

Comments on References

The number and quality of references should be adequate for every aspect of the article. Especially the references [4 - 9] from one issue of a journal must be supported by other references. Booklets, reports, and conference are good to mention, but they might not base on scientific research [1, 2, 10, 12]. The language of the references should be mentioned if they are different from English [1 - 3].

Do not refer relatively to many review article, here 2 out of 16 [9, 13].

Author Response

Thank you for reviewing our manuscript very carefully.

First of all, I want to point out that I appreciate the work and research you did. I was asked to review your manuscript. Due to the number of comments, I chose this chronicle way through the text referring to line numbers, figures and tables. Keep in mind that even enhancing the text by reflecting my comments might lead to new questions and comments.

Response: Thank you for your detailed comments. We respond to your comments as follows.

Comments on the ABSTRACT

Line 12 and 13:

What MRT and SET* stand for? There are NO definitions of this in the whole article. I assumed that you meant mean radiant temperature, abbreviated MRT, or in another version, Tmrt. Probably, with SET* you refer to the Standard Effective Temperature. Please correct this and explain furthermore, why you used SET* instead other thermal comfort indices. Don't forget the references.

Response: We added “mean radiant temperature (MRT)” (Line 12) and “standard new effective temperature (SET*)” (Line 13). And, we added the references regarding outdoor thermal comfort (Lines 119-136).

Line 14/15:

Please add 'more' to '…shading is more effective in…' and 'the' to 'the daytime than the improving the land coverage'.

Response: We added “more” (Line 14) and “the” (Line 15).

Further comments on the Abstract

You should here describe your results in a more common way. To my opinion the sentence starting from 'Even around noon, the pedestrians … ' do not belong here.

Response: We deleted a sentence “Even around noon, the pedestrians …” And, taking into account the opinion by the other reviewer, we added supplement explanation; “The following strategy of extreme high temperature measures is derived by considering the thermal environment map in street canyon.” (Lines 15-17)

Comments on the INTRODUCTION

First of all, I was expecting more information about the thermal conditions of the city of Kobe. By reviewing your research somewhere from the opposite side of Earth, I know less about the climate and the city of Kobe itself. Actually, there was no definition of extreme temperature. Thermal perception also varies in different climates. To understand the thermal severity it might be useful to have information about heatwaves (occurrence and length). Please add some basic climatological information. Also, a short geographical description of Kobe is needed. Kobe, as a city located on sea-side, benefits from the maritime aspects of the climate (sea breeze, humidity). Therefore it is interesting that you found out that in Kobe solar insolation is more dominant on the distribution of SET* as the wind.

Response: We added “Kobe city is located facing Osaka bay. The climate is classified as warm and temperate. According to Köppen and Geiger, this climate is classified as Cfa. The average annual temperature is 16.7 °C. The average annual rainfall is 1,216 mm. Daily maximum and minimum air temperature in Kobe city, from July to September, 2017 to 2019 is shown in Figure 1. In Japan, a day with a minimum air temperature of 25 °C or higher is called a tropical night and used as an index of nighttime sleepiness, and a day with a maximum air temperature of 35 °C or higher is called an extreme hot day and is used as an index of daytime heat. The numbers of tropical nights from 2017 to 2019 are 52, 51, 46 days, and an air conditioner is essential for sleeping. The numbers of extreme hot days from 2017 to 2019 are 3, 12, 7 days, so the last two years were extremely hot.” (Lines 67-75)

Another aspect to understand your research is why this area was chosen as an area of interest (AOI). It seems that the building structure is two-folded, in the westerly part of AOI are according to maps are middle to large size buildings and in the easterly part is smaller to middle size houses. In such a type of research, you performed the contracting authorities normally want to know more about the thermal influences on different types of inhabitants or customers of the AOI. This increases the impact of researches on decision-makers. Are there such aspects of research? If there are, mention them and refer to other studies with references.

Response: We added supplement explanation; “The objective area is the economic, administrative, and cultural center of Kobe city, where people gather from inside and outside the city. Sannomiya station is located in the northeastern end and Motomachi station is located in the northwestern end of this area. The south end of this area is connected to the port area. The east-west shopping streets are located on the north side of this area, but because these pedestrian streets have arcades, they were excluded from the calculation in this study. Relatively large-scale buildings such as offices, department stores, banks, museums, hotels, city halls, and condominiums are located from the center to the south side of this area.” (Lines 147-154)

One word about the term 'land cover (also land coverage)', mentioned, inter alia, in the keywords and line 15. To my understanding, there are a number of researches about the influences of the different land cover pattern on urban micro-climate. Normally they refer to a longer period of investigation under different types of weather. But it is still very difficult to extract the impact of a specific small land cover, especially, if you have only one day with validation measurement. Later in my comments about your validation measurements, I will open-up the issue more.

Response: We also added a note; “The accuracy of the calculation result is compared with the measurement result in Section 3.3.” (Lines 160-161)

Line 27:

The sentence starting in line 27: 'In response to a request …' is very long and at the end quite unclear. Please rephrase it.

Response: We changed this sentence; “In response to a request from the Kobe city local government, the authors presented a proposal for Kobe city policy mainly on the basis of the knowledge of the art. It is constituted by priority introduction places for adaptation measures for extreme high temperature, effects of adaptation measures for extreme high temperature, and hot spots distribution in Kobe city.” (Lines 27-31)

Line 36:

To enhance the relevance of your scientific work you must substantially increase the number of references and refer to them one by one. Here you mentioned six references from ONE issue. This is not enough. Because there is already plenty of research around the world concerning the research topic, you will find easily different opinions and solution.

Response: We added a brief description of each cited reference; “Evaporative cooling effects such as irrigation [4,5], vegetation and pavement watering [5] have been studied by the numerical simulation. Some of those scenarios assumed the future climate affected by climate change [5,6]. Discussions including the improvement of thermal environments in the street canyon or in the plaza were not sufficiently conducted based on the evaluation of the human thermal comfort in previous examinations [7–9].” (Lines 38-43)

Line 45/46:

Here again, you have to open-up the references. Reference [13] is a review article - not research on urban climate maps.

Response: We added supplement explanation; “Many useful examples of large cities such as Tokyo and Beijing, medium-sized cities such as Salvador and Berlin, and small cities such as Sendai and Stuttgart were presented in the book published by Ng and Ren [12]. They contributed to bridging the gap between the science of urban climatology and the practice of urban planning.” (Lines 51-54)

Line 48:

What do you mean with the sentence: 'Spatial distribution …' and especially with the word 'little'?

At the end of the introduction, you should mention at least the definition of the Local Climate Zones by Stewart, I.D. and Oke, T.R. (2012). At least here at the end of the introduction you should define and rephrase your research goals.

Response: We changed “little” to “a little” (Line 58) and added supplement explanation; “The purpose of this study is to derive a strategy of extreme high temperature measures based on the thermal environment map in street canyon. The scale targeted in this study is smaller than the scale generally discussed as the local climate zone [15].” (Lines 62-65)

Comments on chapter 2

Make the header to 'Methods and results', skip 'Calculation'.

Response: We deleted “Calculation” from header (Line 66)

Open-up the SET* thermal index: development, history, use, advantages and possible disadvantages.

Response: We added the description of thermal index including SET; “As Nouri et al. [18] pointed out, the selection of the index for the assessment of outdoor thermal comfort conditions is still a debated matter [19]. They stated that, “So far, within the international community various indices have been developed and disseminated, including the (i) Standard Effective Temperature (SET*) [20]; (ii) Outdoor Standard Effective Temperature (OUT_SET*) [21,22]; (iii) Perceived Temperature (PT) [23]; (iv) Predicted Mean Vote (PMV) [24,25]; (v) Index of Thermal Stress (ITS) [26]; (vi) Predicted Percentage of Dissatisfied (PPD) [24]; (vii) COMFA outdoor thermal comfort model [27]; (viii) Universal Thermal Climate Index (UTCI) [28–31]; (ix) Wet Bulb Globe Temperature (WBGT) [31,32]; and (x) Predicted Heat Strain (PHS) [33–35].” In Japan, SET* and WBGT are mainly used. WBGT, which is a stress index worldwide accepted as a preliminary tool for the assessment of hot thermal environments [36–38], is often used under more severe conditions to warn of the risk of heat stroke. SET* is defined as the equivalent dry bulb temperature of an isothermal environment at 50% RH in which a subject, while wearing clothing standardized for the activity concerned, would have the same heat stress and thermo-regulatory strain as in the actual test environment [20], is used to evaluate the thermal environment [3]. The relationship between SET* and thermal comfort is associated based on the results of a declaration test for the outdoor comfort of Japanese people [39]. SET* is desirable as an index from the viewpoint of appropriately introducing adaptation measures in urban areas and developing a more comfortable outdoor space as it exhibits a good relationship with outdoor thermal comfort [40].” (Lines 119-136)

Line 56 - 58:

This is a good remark of direct solar radiation. But buildings can store energy in the late forenoon hours and radiate it in the early afternoon hours. You might mention this option, too.

Response: We added supplement explanation; “Since long-term calculations are not carried out, the effect of the stored heat of solar radiation in the morning on the surface temperature from afternoon to evening is considered, but the effect of the previous day's effect on the surface temperature in the morning is not considered.” (Lines 91-94)

Line 69:

You were analysing day-time weather condition. Are the wind conditions during day and night the same? Kobe is located on the sea-site. No stronger winds were measured? What about land - and sea-breeze circulations?

Response: We added supplement explanation; “Since the frequency of land breeze decreased due to urbanization, a land breeze with opposite wind direction to sea breeze was sometimes confirmed at night, but the wind direction was almost stable due to the sea breeze during the day.” (Lines 101-103)

Line 78:

On line 78 you used the term '1.5 m high' for the first time. In thermal map research, you are focused on a specific height, here 1.5 metres. This height could stand for the centre of the human corpus. Other thermal indices like PET refer to 1.75 metres as height and therefore as a parameter to calculate the body surface. Please switch the indication of size from 'high' to 'height'.

Response: We changed “high” to “height” (Line 108).

Figure 2:

Compass rose missing. Legend missing. There are 8 pairs of measurement points (red dots), what are the other green dots standing for?

Response: We added the compass, arranged legend “wind velocity (m/s)” and added supplementary explanation “(16 pairs)”. (Line 117)

The wind floating scale of 18 classes with wind maximum of 3.6 metres/second out-site the AOI is nice but not very effective, you cannot see any differences.

Response: We added supplement explanation; “Although 4.3 m/s is given to the upper wind velocity, the wind velocity on the pedestrian level is high only in the windward waterfront area and is less than 1 m/s in 90.6 % of the objective area, due to the resistance by the middle-rise buildings.” (Lines 110-112)

Line 84/85:

Red circles: You did not mention how many measurement points you had, how did you choose them, and where they were located.

Response: We added supplement explanation; “16 pairs of measurement points are set on the sidewalks on both sides of the east-west and north-south streets with different road widths.” (Lines 199-200)

Line 91:

What is the definition of 'Building shape'?

Response: We added supplementary explanation “(gray top view)” (Line 163).

Line 94:

We normally use 'grid' instead of 'mesh'. Correct this from here on.

Response: We changed “mesh” to “grid” (Line 154).

Line 97:

How many trees are in the AOI?

Response: We added supplement explanation; “Ratio of tree canopy area in the entire objective area is 9.7 %, which is almost the same in the east-west road and the north-south road. It is large at 31.9 % in central park and small in intersection and open space.” (Lines 158-160)

Figure 3:

Compass rose missing (important, because the figure is slightly turned). Typo in the legend: Block (white).

Response: We added the compass and changed the legend. (Line 162)

Figure 4:

Compass rose missing (important, because the figure is slightly turned).

I suggest a different title: Classification of open space in the study area.

Response: We added the compass and changed the title (Lines 164-165).

Figure 6:

Compass rose missing (important, because the figure is slightly turned).

Legend missing.

And again an addition to the title: trees in the study area. To support the measurements results, it would be nice to indicate the sites of the measurements (red dots) here, too.

Response: We added the compass, legend and measurement points, and changed the title (Lines 168-169).

Comments on chapter 3

Line 112 - 114:

If you decide to define SET*, then this sentence is not necessary any more.

Response: We left the explanation for the readers’ understanding (Lines 176-178).

Figure 7 and 8:

Compass rose missing (important, because the figure is slightly turned).

As mentioned you have 8 measurement points (!), you did not mention the measuring height at these dots. Furthermore, did you use recording devices for measurements? Where the devices hand-held or stand-alone? How many thermal pictures were taken and how? Unfortunately, you don't have and measurements in the open park area, which make references of SET* and MRT to this are not proved.

Response: We added the compass and the information about measurement points “16 pairs” (Line 199) and height “at 1.5 m height” (Line 202). The outline of the measurement method is described as follows; “Two persons moved each measurement point during the above period and measured them after staying at each measurement point for a while. Air temperature and relative humidity were measured at 1.5 m height by a thermistor and a capacitance sensor inside a ventilation system with a shade. Wind velocity was measured by a hot-wire anemometer after confirming the wind direction by a windsock. Surface temperature on the surrounding ground and wall surfaces was measured by an infrared thermometer, which was supplemented by an infrared image by a thermal camera. A fisheye photograph was taken at each measurement point, then MRT and SET* were calculated, given the same assumptions as the above calculation methods.” (Lines 200-208)

Line 138 - 140:

Add 'in the year 2019' to the end of the sentence, because you cannot generalised from two days measurements, that you have a typical overall summer day.

Response: We added “in the year 2019” (Line 211).

Figure 9:

Generally speaking, you were measuring neither MRT nor SET*, both are calculated from air temperature, humidity, radiation and wind, as you should describe earlier. The MRT and SET* interrelationship is well seen. But as you are interested in extreme high temperature an additional figure with these really measured air temperatures would be here a useful complement.

Response: We added supplement explanation; “Since it took about two hours for mobile measurements, the spatial difference was indistinguishable from the temporal difference in air temperature and relative humidity measurement results. On the other hand, a clear difference was confirmed in the spatial distribution of MRT based on the sun shade and the surface temperature distribution. Although slight difference was also confirmed in the spatial distribution of wind velocity affected by surrounding features, the influence of the MRT was dominant for the spatial distribution of SET*.” (Lines 215-221)

Comments on chapter 4

You should mention here that the discussion is based on modelled data.

Line 158:

What do you mean by 'improving the land coverage'? Isn't the increase of shading (e.g. by trees or other obstacles) an improving of the land coverage? I don't see here the controversial aspect.

Response: We changed “improving the land coverage” to “the difference of land cover” (Lines 234-235).

Figure 10, 11, 12:

Add ' Modelled' to the title.

Response: We added “modeled” to the title (Lines 237, 253, 256).

Line 185 - 186:

How this dominance is expressed? Is it not possible just to say that sunny places are generally warmer (hotter) than shaded places in all widths of street canyons?

Response: We changed the sentence to “Sunny places are generally warmer (hotter) than shaded places in all widths of street canyons.” (Lines 262-263)

Figure 13 - 15:

What are the whiskers in the figures stands for? Are they needed?

How do explain the low variability of SET* of all street canyon types and widths? For sunny and shaded sites the differences are within 2 degrees of SET* and still in the discomfort zone of SET*.

Response: We added supplement explanation; “The plot is the averaged value and the vertical bar on each plot is the standard deviation in the street width section.” (Lines 267-268, 271-272, 275-276) and added supplement explanation; “In all three roads, if the street is divided into sunny and shade places, the variation of SET* is small because the influence of solar radiation is large.” (Lines 263-264)

Comments on chapter 5

Line 197:

Because you modelled SET* and MRT on one day and measured the air temperature, humidity, thermal building conditions, and wind twice in two-hour sessions for the selected sites on that day, this is not enough to classify this day as typical summer day without any validation with long-term climate data. Please add this as a notice, that your research is based on day, August, 5th, 2019, or validate your measurements against climate data.

Then you can conclude that MRT is more dominant than wind velocity on SET*.

Response: We added the note; “However, this paper presents the analysis results only on a specific typical sunny summer day. It represents a period called the extreme hot day in Japan, but additional analysis is needed for cloudy days and days with different solar altitudes.” (Lines 282-284)

Comments on References

The number and quality of references should be adequate for every aspect of the article. Especially the references [4 - 9] from one issue of a journal must be supported by other references. Booklets, reports, and conference are good to mention, but they might not base on scientific research [1, 2, 10, 12]. The language of the references should be mentioned if they are different from English [1 - 3].

Do not refer relatively to many review article, here 2 out of 16 [9, 13].

Response: We added some significant references [12, 15, 18-38, 40], added a brief description of each cited references [4-9], left information about social interests and efforts in this area [1-3,10, 13], added “(in Japanese)” to reference [1-3, 39] and added explanation for supplementing review reference [9, 14].
